# LEARNING TO GENERATE BETTER THAN YOUR LLM

## ABSTRACT

Reinforcement learning (RL) has emerged as a powerful paradigm for fine-tuning Large Language Models (LLMs) for text generation. In particular, recent LLMs such as ChatGPT and GPT-4 can engage in fluent conversations with users after finetuning with RL. Inspired by *learning-to-search* algorithms and capitalizing on key properties of text generation, we seek to investigate RL algorithms beyond general purpose algorithms like Proximal Policy Optimization (PPO). In particular, we extend RL algorithms to allow them to interact with a dynamic black-box guide LLM and propose RL *with guided feedback* (RLGF), a suite of RL algorithms for LLM fine-tuning. We experiment on the IMDB positive sentiment, CommonGen, and TL;DR summarization tasks. We show that our RL algorithms achieve higher performance than supervised learning (SL) and RL baselines, demonstrating the benefit of interaction with the guide LLM. On both CommonGen and TL;DR, we not only outperform our SL baselines but also improve upon PPO across a variety of metrics beyond the one we optimized for.

## 1 INTRODUCTION

Large Language Models (LLMs) have become very capable in various real-world applications ranging from being able to answer open-ended questions on numerous topics (Zhang et al., 2022), write articles from short descriptions (Goyal et al., 2022), generate code (Github, 2023), follow robot commands (Huang et al., 2022), solve puzzles (Bubeck et al., 2023), and even showcased as assistive models for education (Khan Academy, 2023) and healthcare (Lee et al., 2023b).

However, using supervised learning (SL) to train LLMs presents a challenging metric mismatch (Wiseman & Rush, 2016) between the training and testing regimes. The metric mismatch arises from the training metric being the log-loss while the testing metrics are task-specific such as BLEU or user satisfaction rating. This discrepancy is magnified when fine-tuning LLMs on downstream tasks where the main goal is not just producing fluent text but also being proficient at solving the specific task.

Reinforcement Learning (RL) by definition address this metric mismatch by directly optimizing the metrics through reward feedback. Recently, OpenAI fine-tuned LLMs with RL from human feedback (RLHF) to better align LLMs to human intentions, leading to the great success of ChatGPT (OpenAI, 2023). Recently, GRUE benchmark (Ramamurthy et al., 2022) systematically studied RL versus SL when finetuning LLMs on downstream tasks with predefined rewards. GRUE's preliminary results demonstrate the benefit of RL when fine-tuning LLMs, leading to the release of popular codebases such as RL4LMs (Ramamurthy et al., 2022), TRLx (CarperAI, 2023) and AlpacaFarm (Dubois et al., 2023), that enables RL for language models. However, ChatGPT, RL4LMs, TRLX, and AlpacaFarm all use vanilla policy gradient methods known to be sample inefficient and sensitive to local minima due to the combinatorially large search space of natural language generation (Ramamurthy et al., 2022).

In this work, we focus on more efficient ways of fine-tuning LLMs on downstream tasks with predefined rewards. Our approach is motivated by prior work on Imitation Learning (IL) for structured prediction, which often leverages an existing guide policy (not necessarily an optimal policy) to reduce the search space for more efficient and optimal learning. Our key observation is that since modern LLMs exhibit impressive general language capabilities, they can serve as guide policies to improve the RL procedure. Our framework, which we call, *RL with guided feedback* (RLGF), integrates a guide policy into a policy gradient framework. The guide policy can provide reasonable but sub-optimal predictions for downstream tasks, which our framework can then leverage to learn

a near-optimal strategy. We introduce novel algorithms for fine-tuning LLMs using our RLGF framework while capturing various existing IL for structured prediction and RL algorithms.

We evaluate on three tasks. The first is IMDB where the goal is to generate a positive and fluent review given an initial context. The second is CommonGen where the goal is to write a fluent text that uses a given set of words. Finally, we test on the TL;DR summarization task where the objective is to learn to generate summaries using human preference data. For all tasks, we find evidence of metric mismatch from SL-based fine-tuning approaches and show that RL-based methods which utilize reward signals outperforms on the task metric. We then demonstrate RLGF outperforming PPO on reward, fluency, as well as automated lexical metrics such as Rouge. Finally, we investigate how various baselines and RLGF algorithms balance the inherent trade-off between reward optimization and the KL constraint in the RLHF objective. We provide both theoretical justification and empirical evidence to show the benefit of using feedback in RL for fine-tuning LLMs on downstream tasks.

## 2 RELATED WORK

Here we present the most relevant works at the intersection of IL, RL, and natural language generation. Please see Appendix A for a more thorough treatment of the literature.

**IL for Structured Prediction:** Algorithms such as Schedule Sampling (SS) (Bengio et al., 2015), methods using SS (Duckworth et al., 2019; Mihaylova & Martins, 2019; Goyal et al., 2017), SEARNN (Leblond et al., 2017), Bridging the Gap (Zhang et al., 2019b), Mixer (Ranzato et al., 2015) been inspired by IL for structured prediction algorithms DAGGER (Ross et al., 2011), DAD (Venkatraman et al., 2015), and SEARN (Daumé et al., 2009). Our work is inspired by AggreVaTeD (Sun et al., 2017) (Differentiable AggreVaTe Ross & Bagnell (2014)) where the algorithm makes use of differentiable policies and multi-step feedback rather than immediate one-step predictions to imitate. Similarly, we present a differentiable version of LOLS (Chang et al., 2015) as well as an improvement, $D^2$LOLS.

**LLM Fine-tuning from Human Preferences:** Recent advancements in fine-tuning of Large Language Models (LLMs) have shown incredible success in tasks through learning from human preferences. Being simpler to accumulate human preferences, Reinforcement Learning from Human Feedback (RLHF) (Stiennon et al., 2020) introduced a paradigm to utilize RL to improve downstream performance on translation (Kreutzer et al., 2018b), summarization (Stiennon et al., 2020), storytelling (Ziegler et al., 2019), and instruction following (OpenAI, 2023). Although effective, following works have shown RLHF to be challenging due to reward hacking, difficulties in scaling, and training instability (Zhao et al., 2023; Rafailov et al., 2023; Liu et al., 2023). To circumvent these difficulties, recent works have proposed methods to optimize for human preferences without RL (Zhao et al., 2023; Yuan et al., 2023; Rafailov et al., 2023; Liu et al., 2023). DPO, SLiC, RRHF, and RSO are methods that optimize for compatibility with a preference dataset under a preference reward model such as the Bradley Terry model (Bradley & Terry, 1952). In contrast, our work takes a different approach to improving RLHF by investigating improvements to PPO (Schulman et al., 2017), the base RL algorithm used.

**LLM Distillation:** With an ever growing arsenal of powerful, black-box LLMs, recent work has aimed to distill specific capabilities into a smaller model. Knowledge distillation (Buciluǎ et al., 2006; Hinton et al., 2015) in autoregressive models investigated matching sequence level log probabilities (Kim & Rush, 2016), model hidden states (Jiao et al., 2019), or attention scores (Wang et al., 2020). Recently, more sophisticated methods, inspired from the IL literature, are being proposed to better imitate the expert LLM's performance (Lin et al., 2020a; Agarwal et al., 2023; Mukherjee et al., 2023), with ORCA (Mukherjee et al., 2023) reaching parity performance with ChatGPT (OpenAI, 2023) by distilling the reasoning traces from GPT4 (OpenAI, 2023). Distinct from this line of work, RLGF does not aim to replicate the guidance policy. Rather, our objective is to leverage generation traces derived from a guide policy to condense the search space for RL algorithms. More importantly, our goal goes beyond imitation of the guidance policy and focuses on algorithms that better optimize a reward with guidance policy feedback.

## 3 PRELIMINARIES

Text generation with LLMs can be viewed as a structured prediction problem, consisting of an input space $\mathcal{X}$, an output space $\mathcal{Y}$ and non-negative loss function $\ell(x, \hat{y}, y*) \mapsto \mathbb{R}^{\geq 0}$ such that the loss

function $\ell$ represents how close $\hat{y}$ is to the ground truth $y^*$ given the input $x$. We are provided with a training set of $N$ labeled input-output pairs $\mathcal{D} = \{(x^i, y^i)\}_{i=1}^N$ drawn from some unknown distribution over $\mathcal{X} \times \mathcal{Y}$. The goal is to learn a mapping $f : \mathcal{X} \mapsto \mathcal{Y}$ that minimizes the loss function $\ell$ with respect to $\mathcal{D}$. We adopt the approach of solving the text generation structured prediction problems using sequential decision-making as formalized in learning-to-search (L2S) (Daumé et al., 2009; Collins & Roark, 2004; Ratnaparkhi, 1996).

We view our L2S problem as a token-level finite-horizon MDP $\langle \mathcal{S}, \mathcal{A}, P, R, H, \mu \rangle$ using a finite vocabulary $\mathcal{V}$. We are given a labeled dataset $\mathcal{D} = \{(x^i, y^i)\}_{i=1}^N$ of $N$ samples, where $x^i$ is a prompt text and $y^i$ is the target text generation. We define $\mu \in \Delta(\mathcal{D})$ as the initial distribution over prompts in the dataset, and the action space $\mathcal{A}$ as the set of tokens in our vocabulary $\mathcal{V}$. The state space $\mathcal{S} = \cup_{h=1,\cdots,H} \mathcal{V}^h$ is the set of all possible token sequences and a state $s_h \in \mathcal{S}$ is the prompt $x$ and previously generated tokens $(a_0, a_1, \ldots, a_{h-1})$, i.e., $s_h = (x, a_0, a_1, \ldots, a_{h-1})$. The transition function $P : \mathcal{S} \times \mathcal{A} \to \Delta(\mathcal{S})$ is a deterministic known transition function that appends the next action $a_h$ to the state $s_{h+1}$ The time horizon $H \in \mathbb{Z}_+$ is the maximum generation length. Finally, $R : \mathcal{S} \to \mathbb{R}$ is the reward function such as the task evaluation metric.

Let $d_h^\pi$ represent the state distribution of visiting a state at time $h$. Let $d^\pi = \frac{1}{H} \sum_{h=0}^H d_h^\pi$ be the average visitation if we follow $\pi$ for $H$ steps in a trajectory. With an LLM policy $\pi$, we define the value function and $Q$-function as $V_h^\pi(s) = \mathbb{E}_\pi[\sum_{h'=h}^H R(s_{h'})|s_h = s]$ and $Q_h^\pi(s, a) = R(s) + \mathbb{E}_{s' \sim P(\cdot|s,a)}[V_{h+1}^\pi(s')]$ respectively. Finally, we define the advantage function for an LLM policy $\pi$ as $A^\pi(s, a) = Q^\pi(s, a) - V^\pi(s)$.

**Guide policy $\pi^g$** In our setting, we additionally assume access to an LLM guide policy $\pi^g$ that can assist our policy $\pi$. The guide policy can be used to alter the initial state distribution $\mu$ and to compute the advantage function $A^{\pi^g}(s, a)$. In this work, $\pi^g$ is a supervised fine-tuned (SFT) model on the downstream task and generate feedback from $\pi^g$ with a more effective decoding strategy like nucleus sampling (Holtzman et al., 2019). Note, RLGF treats $\pi^g$ as a query-able, black-box model that we cannot update. This allows for $\pi^g$ to be any black-box model such as GPT4 or a human-expert.

## 4 REINFORCEMENT LEARNING FROM GUIDED FEEDBACK

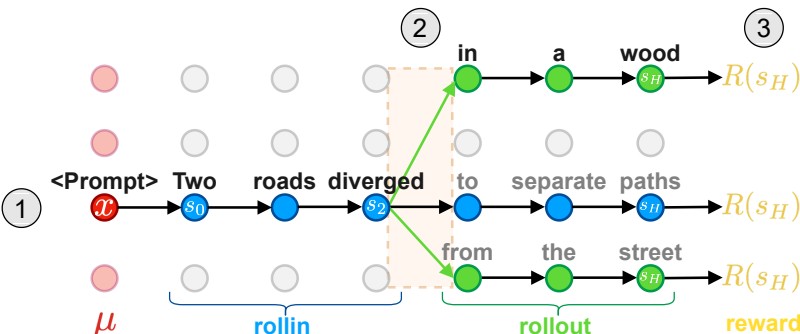

Figure 1: RLGF's main mechanism of incorporating guidance through interactions between two LLMs: rollin and rollout policies. (1) the rollin policy generates a trajectory. (2) the rollout policy restarts to a sampled point in the generation (i.e. $s_2$) and completes the generation. (3) the rollout policy receives a score (i.e. reward) for the generation.

Unlike other tasks studied in RL, structured prediction problems such as text generation, have two key properties: a deterministic transition function and a policy's ability to restart to any state. Because our transition function is the set of previously generated tokens, we can easily alter the words in the generation (add, remove or swap), and restart our policy $\pi_\theta$ to any point of the generation.

Restarts allow us to execute rollin and rollout policies as seen in Figure 1. The rollin policy is used to generate sequences that the rollout policy evaluates. Specifically, we sample a prompt $x$ and target sentence $y$ from our initial distribution $\mu$. We then generate an entire trajectory using our rollin policy starting from the sampled prompt. We combine the state-action pairs from the collected rollin

trajectory with the initial state distribution – creating a modified initial state for the rollout policy. The rollout policy samples a state along the rollin generation, restarts to this state and performs a one-step deviation action. The rollout policy then completes the generation and collects a reward. The rollin and rollout policies can be our LLM policy $\pi_\theta$, guide policy $\pi^g$ or a mixture that interpolates between the two. Depending on the choice of rollin and rollout policies, we invoke different algorithms.

**PPO: Rollin $\pi_\theta$ and Rollout $\pi_\theta$** Under this schematic, notice how when both the rollin and rollout policies are our current LLM policy $\pi_\theta$ that is being fine-tuned, the resulting RL algorithm is PPO. That is, we would be collecting generations from a single LLM. This configuration does not take advantage of the ability to modify the initial state distribution nor the availability of a guide policy $\pi^g$.

---

**Algorithm 1** PPO$^{++}$

---

1: **Input:** $\pi_\theta$, guide $\pi^g$, iterations $T$, mixing parameter $\beta \in [0, 1]$, dataset $\mathcal{D} = \left\{(x^i, y^i)\right\}_{i=1}^N$
2: **for** $t \in [T]$ **do**
3:     Rollin with $(s, a) \sim \beta d^{\pi^g} + (1 - \beta) d^{\pi_\theta^t}$ starting from $x \sim \mathcal{D}$
4:     Rollout with $\pi_\theta^t$ to collect trajectories
5:     Update $V_\phi^{\pi_\theta^t}$ with trajectories and compute advantage estimates $A^{\pi_\theta^t}$
6:     Update $\pi_\theta$ using PPO loss with $A^{\pi_\theta^t}$
7: **return** $\pi_\theta$

---

**PPO$^{++}$: Rollin $\pi^g$ and Rollout $\pi_\theta$** The second scheme we consider is rollin with our guide policy $\pi^g$ and rollout with our LLM policy $\pi_\theta$. This strategy is motivated from a popular Approximate Policy Iteration algorithm (Bertsekas, 2011): Conservative Policy Iteration (CPI) (Kakade & Langford, 2002). CPI proposes to use a diverse initial state distribution to address the exploration issue in PG methods. Particularly, it proposes to use an initial state distribution that covers some high-quality policy distribution. The first key idea of PPO$^{++}$ is to take advantage of a guide policy $\pi^g$ to provide an enlarged initial state distribution – so that the rollout policy, $\pi_\theta$, can visit diverse and relevant states it would otherwise not visit. The second key idea of PPO$^{++}$ is using a mixture policy with state distribution $\beta d^{\pi^g} + (1 - \beta) d^{\pi_\theta}$, for rollin (see Algorithm 1 Line 3). This ensures that with probability $(1 - \beta)$, PPO$^{++}$ is executing the default PPO update, making sure PPO$^{++}$ never underperforms PPO.

---

**Algorithm 2** AggreVaTeD

---

1: **Input:** $\pi_\theta$, guide $\pi^g$, iterations $T$, mixing parameter $\beta \in [0, 1]$, dataset $\mathcal{D} = \left\{(x^i, y^i)\right\}_{i=1}^N$
2: **for** $t \in [T]$ **do**
3:     Rollin with $(s, a) \sim (1 - \beta) d^{\pi_\theta^t} + \beta d^{\pi^g}$ starting from $x \sim \mathcal{D}$
4:     Rollout with $\pi^g$ to collect trajectories
5:     Update $V_\phi^{\pi^g}$ with trajectories and compute advantage estimates $A^{\pi^g}$
6:     Update $\pi_\theta$ using PPO loss with $A^{\pi^g}$
7: **return** $\pi_\theta$

---

**AggreVaTeD: Rollin $\pi_\theta$ and Rollout $\pi^g$** The next scheme performs rollin with our LLM policy $\pi_\theta$ and rollout with our guide policy $\pi^g$ – the opposite of PPO$^{++}$. This scheme is an interactive imitation learning algorithm, AggreVaTeD (Sun et al., 2017), a differentiable policy gradient version of AggreVaTe (Aggregate Values to Imitate (Ross & Bagnell, 2014)) as seen in Algorithm 2. AggreVaTeD is an API algorithm similar to CPI and also uses a mixture policy with state distribution $\beta d^{\pi^g} + (1 - \beta) d^{\pi_\theta}$ for rollin. This algorithm first generates rollins with the mixture policy to collect sequences. Then AggreVaTeD generates rollouts with the guide policy and evaluates the quality of the generated rollouts. It then uses the rollouts to train a value network $V_\phi^{\pi^g}$ that measures the reward-to-go of $\pi^g$, which in turn is used to construct the advantage of $\pi^g$: $A^{\pi^g}$. With this advantage $A^{\pi^g}$, AggreVaTeD updates the policy like PPO. Intuitively, the algorithm aims to learn the policy $\arg\max_a A^{\pi^g}(s, a)$. Rolling out with $\pi^g$ ensures that the LLM policy $\pi_\theta$ can be *at least* as good as or better than the guide policy $\pi^g$.

---

**Algorithm 3** D$^2$LOLS

---

1: **Input:** $\pi_\theta$, guide $\pi^g$, iterations $T$, dataset $\mathcal{D} = \left\{ (x^i, y^i) \right\}_{i=1}^N$
2: Run $\pi_\theta^1 = \texttt{AggreVaTeD}(\pi_\theta, \pi^g, \alpha T, \beta_1, \mathcal{D})$
3: Run $\pi_\theta^2 = \texttt{PPO}^{++}(\pi_\theta^1, \pi^g, (1-\alpha)T, \beta_2, \mathcal{D})$
4: **return** $\pi_\theta^2$

---

**D$^2$LOLS: combines PPO$^{++}$ and AggreVaTeD**  Given the previous approaches of interaction, we can come up with multiple ways to combine PPO, PPO$^{++}$, and AggreVaTeD. In Algorithm 3, we present Direct and Differentiable Locally Optimal Learning to Search (D$^2$LOLS), which is a simple approach to combine the previous methods. D$^2$LOLS is a differentiable policy gradient version of Locally Optimal Learning to Search (LOLS)(Chang et al., 2015) and addresses limitations of how LOLS combines PPO, PPO$^{++}$, and AggreVaTeD. The original formulation of LOLS requires computing cost-sensitive classification similar to AggreVaTe; instead we take inspiration from AggreVaTeD's differentiable approach to develop a differentiable version of LOLS. Furthermore, LOLS (Algorithm 4) has a mixing probability parameter $\alpha$ which directly merges the advantage function between PPO and AggreVaTeD, leading to theoretical issues. D$^2$LOLS removes this mixing probability and replaces it with a mixing time variable $\alpha$ that decides how many iterations to perform AggreVaTeD before switching to PPO$^{++}$. This simple strategy fixes LOLS's issue arising from interweaving guidance.

## 5 THEORETICAL JUSTIFICATION

In this section, we provide theoretical justification for various rollin and rollout schemes mentioned in Section 4. Each algorithmic scheme takes advantage of a guide policy $\pi^g$, the ability to restart the policy to any state, and access to the reward signal. Our theoretical justification are derived from the original algorithms that each method has built upon.

**Interactive Imitation Learning: AggreVaTeD**  In our interactive IL setting, we assume access to the ground truth reward and to a guide policy $\pi^g$ that may not necessarily be an expert policy $\pi^\star$ (i.e. optimal at the task). Our AggreVaTeD (Algorithm 2) implementation is a modification of the original AggreVaTeD (Sun et al., 2017) to incorporate a PPO policy gradient loss. The overall idea is to perform policy gradient updates on the loss function $\ell_t(\pi) := \mathbb{E}_{s \sim d^{\pi^t}} \mathbb{E}_{a \sim \pi(\cdot|s)}[A^{\pi^g}(s, a)]$, where $\pi^t$ is our latest learned policy. We can define the average-regret and best policy performance in our policy class over $T$-iterations as:

$$\epsilon_{\text{regret}} = \frac{1}{T} \left( -\sum_{t=0}^{T} \ell_t(\pi^t) + \max_{\pi \in \Pi} \sum_{t=0}^{T} \ell_t(\pi) \right) \quad \epsilon_{\text{class}} = \max_{\pi \in \Pi} \frac{1}{T} \sum_{t=0}^{T} \mathbb{E}_{s \sim d^{\pi^t}} \left[ A^{\pi^g}(s, \pi(s)) \right].$$

If the gradient update procedure achieves no-regret, i.e., $\epsilon_{\text{regret}} \to 0$ as $T \to \infty$, AggreVaTeD achieves the following guarantee; there exists $t \in [T]$, such that:

$$V^{\pi^t} \geq V^{\pi^g} + H\epsilon_{\text{class}}.$$

When the guide policy is included in our policy class $\pi^g \in \Pi$, e.g., when our policy $\pi_\theta$ and our guide $\pi^g$ have the same GPT2 model architecture, then our $\epsilon_{\text{class}}$ term is guaranteed to be non-negative. Furthermore, this term is positive when $\pi^g$ is not globally optimal with respect to its advantage function (i.e., $\max_a A^{\pi^g}(s, a)$ can be positive). Thus when $\epsilon_{\text{regret}} \to 0$ (i.e., no-regret), AggreVaTeD guarantees to learn a policy $\pi_t$ that outperforms the guide policy by a margin. This was originally confirmed empirically in Sun et al. (2017) and is also confirmed in our experiments. With our SFT model with nucleus sampling as $\pi^g$, AggreVaTeD learns a policy $\pi^t$ outperforming $\pi^g$.

**Reinforcement Learning with better restart distribution: PPO$^{++}$**  Although AggreVaTeD is capable of outperforming $\pi^g$, it is an imitation learning algorithm, meaning by design, its performance is limited by the performance of $\pi^g$. In contrast, RL has the potential to learn the near optimal policy, but popular RL approaches suffer from a lack of exploration. We propose to leverage rollin's with the guide policy to overcome RL's exploration issues. PPO$^{++}$ Algorithm 1 implements this idea using a PPO loss. We can interpret the rollin policy distribution with the guide policy, as a restart

distribution that alters the initial distribution of our policy, i.e., $\mu_{\text{mix}} := (1 - \beta)\mu + \beta d^{\pi^g}$, where recall $\mu \in \Delta(\mathcal{D})$ is the original initial state distribution over our data.

Policy gradient theory (Kakade & Langford, 2002; Bagnell et al., 2003; Agarwal et al., 2019; 2021) ensures that as long as a near optimal policy is covered by the restart distribution, we can learn to perform as well as the near optimal policy. More formally, consider the special case where $\beta = 1/2$, and $\pi^\star$ is the globally optimal policy; and assume that at some iteration $t$ one-step local improvement over $\pi^t$ is small, i.e., $\mathbb{E}_{s,a \sim d^{\pi^t}_{\mu_{\text{mix}}}} \left[ \max_a A^{\pi^t}(s,a) \right] \leq \epsilon$, then with some small $\epsilon$ we have:

$$V^{\pi^t} \geq V^{\pi^\star} - O\left( H^2 \max_s \left( \frac{d^{\pi^\star}(s)}{d^{\pi^g}(s)} \right) \epsilon \right)$$

We refer readers to the proof of theorem 6.2 in Kakade & Langford (2002). Note that compared to the result from `AggreVaTeD`, we are able to compare against the globally optimal policy $\pi^\star$ under the condition that $\pi^g$'s state distribution covers $\pi^\star$'s state distribution (i.e., the guide policy has a good sense of what states $\pi^\star$ will likely visit). In our experiments, we mainly use a SFT model with nucleus sampling as our guide policy $\pi^g$. While we do not expect the SFT policy $\pi^g$ is as good as the optimal $\pi^\star$, it is reasonable to expect that $d^{\pi^g}$ provides coverage to $d^{\pi^\star}$. Our experiments verify that restarting based on states from $d^{\pi^g}$ improves the performance of PPO.

**Combine Reinforcement Learning and Imitation Learning: $\text{D}^2\text{LOLS}$**   $\text{D}^2\text{LOLS}$ is the simplest approach to combine `AggreVaTeD` and $\text{PPO}^{++}$. This algorithm runs `AggreVaTeD` for a fixed period of time and then $\text{PPO}^{++}$ for the remaining time. If our policy gradient algorithm is Trust-region policy optimization (TRPO) [1] (Schulman et al., 2015) or CPI (Kakade & Langford, 2002), then our algorithm has a guaranteed monotonic policy improvement. This means that upon convergence, we achieve two properties: (1) our learned policy is at least as good or better than the guide policy $\pi^g$, (2) our policy is locally optimal, i.e., the local one-step improvement, $\mathbb{E}_{s,a \sim d^{\pi}_{\mu_{\text{mix}}}} \left[ \max_a A^{\pi}(s,a) \right]$, has to be small (otherwise TRPO and CPI can keep improving).

There exist several algorithms in the literature that combine RL and IL (Cheng et al., 2018; Sun et al., 2018; Chang et al., 2015; Rajeswaran et al., 2017; Nair et al., 2018). The key difference between $\text{D}^2\text{LOLS}$ and `LOLS` is how $\text{PPO}^{++}$ and `AggreVaTeD` is combined. `LOLS` uses a mixing probability $\alpha$ to combine our $\pi_\theta$ and the guide policy $\pi^g$ advantage function $\alpha A^{\pi^t_\theta} + (1-\alpha)A^{\pi^g}(s,a)$; whereas $\text{D}^2\text{LOLS}$ uses a mixing time parameter $\alpha$ to decide when to switch from doing `AggreVaTeD` to $\text{PPO}^{++}$ for the remainder of training. `LOLS` can achieve the property of outperforming better than $\pi^g$ and also being locally optimal, but *only under* the assumption that the following gap is small:

$$\forall \pi : \left| \mathbb{E}_{s \sim d^\pi} \left[ \max_a A^{\pi^g}(s,a) + \max_a A^{\pi}(s,a) \right] - \mathbb{E}_{s \sim d^\pi} \max_a \left[ A^{\pi^g}(s,a) + A^{\pi}(s,a) \right] \right| \leq \varepsilon,$$

with some small $\varepsilon$. However, such a gap can exist in practice and does not vanish even with enough training data. Intuitively this gap is non-trivial when the one-step improvement over $\pi$ contradicts with the one-step improvement over $\pi^g$. The simplest approach $\text{D}^2\text{LOLS}$ works the best, and achieves the guarantee that `LOLS` aimed for without the additional assumption of the above gap being small.

## 6 EXPERIMENTS

We perform all of our experiments using a modified PPO objective $J_{ppo}$ (Ouyang et al., 2022; Wu et al., 2016). This objective combines the original PPO objective with a maximum-likelihood estimation (MLE) objective of the ground-truth dataset's $\mathcal{D}$ references:

$$J_{ppo}(\pi) = \mathbb{E}_{(s,a) \sim \pi_\theta} \left[ R(s) - \lambda \text{KL}(\pi_\theta(a|s)||\pi_0(a|s)) \right] + \eta \mathbb{E}_{(s,a) \sim \mathcal{D}} \left[ \log \pi_\theta(a|s) \right],$$

where $\lambda$ is the KL coefficient and $\eta$ is the MLE coefficient. For all of our proposed RLGF algorithms discussed in section 4 we consider setting $\pi^g$ to the supervised fine-tuned model (SFT) with nucleus

---

[1] in our experiments, instead of using TRPO, we use PPO – a scalable version of TRPO that is more suitable for high-dimensional problems. However we emphasize the TRPO and PPO use the same principle for policy optimization: make conservative policy update (Kakade & Langford, 2002) to ensure monotonic improvement.

| Algorithms | IMDB Sentiment *Semantic and Fluency Metrics* | | | CommonGen *Lexical and Semantic Metrics* | | | |
|---|---|---|---|---|---|---|---|
| | **Sentiment Score** ($\uparrow$) | Perplexity ($\downarrow$) | Output-Perplexity ($\downarrow$) | Bleu-4 ($\uparrow$) | BERTScore ($\uparrow$) | **CIDEr-D** ($\uparrow$) | **SPICE** ($\uparrow$) |
| Zero-Shot | $0.48 \pm 0.00$ | $32.55 \pm 0.00$ | $5.64 \pm 0.00$ | 0.16 | 0.93 | 1.10 | 0.26 |
| SFT | $0.55 \pm 0.00$ | $35.67 \pm 0.00$ | $6.19 \pm 0.00$ | 0.22 | 0.95 | 1.43 | 0.31 |
| SFT+PPO | $0.97 \pm 0.01$ | $44.92 \pm 1.78$ | $3.17 \pm 0.62$ | 0.26 | 0.95 | 1.65 | 0.32 |
| SFT+PPO$^{++}$ | $0.97 \pm 0.01$ | $44.83 \pm 2.10$ | $3.34 \pm 0.80$ | 0.27 | 0.95 | 1.68 | 0.32 |
| SFT+AggreVaTeD | $0.95 \pm 0.03$ | $52.56 \pm 5.38$ | $5.04 \pm 2.30$ | 0.27 | 0.95 | 1.65 | 0.32 |
| SFT+LOLS | $0.93 \pm 0.05$ | $53.30 \pm 16.70$ | $3.44 \pm 4.96$ | 0.26 | 0.95 | 1.66 | 0.32 |
| SFT+D$^2$LOLS | $0.97 \pm 0.00$ | $43.88 \pm 2.37$ | $2.92 \pm 0.13$ | 0.27 | 0.95 | 1.69 | 0.33 |

Table 1: **IMDB and CommonGen Results:** We compute the mean and standard deviation over 3 seeds for the IMDB task and compute 1 seed for the CommonGen task. For our reward function each task we use the bold metric(s). The zero-shot model is the performance of the pretrained model used for IMDB and CommonGen, GPT-2 and T5 respectively. SFT+Alg indicates running Alg after supervised finetuning. SFT+nucleus is used as our guide policy $\pi^g$ for all experiments.

sampling for decoding (i.e., $\pi^g$ =SFT+nucleus). We treat SFT+nucleus as a black-box model that we can only query for text generation and do not perform updates to it. By using SFT+nucleus as our guide policy, we run all of our experiments under the exact same conditions as those of RLHF. Note, RLHF already requires keeping SFT to compute the KL constraint, $KL(\pi_\theta || \pi_0)$, in $J_{ppo}$.

**Task Details**  In our experiments, *perplexity* measures how likely our learned model, $\pi_\theta$, is to generate the references in the task dataset, whereas *output perplexity* computes how likely a general LLM (e.g. GPTJ) is to generate the generations from our learned policy, $\pi_\theta$. Both perplexity metrics have been reported as a measure of fluency (Fedus et al., 2018; Ramamurthy et al., 2022).

We perform experiments on three tasks. IMDB is the first task and the objective is to generate fluent and positively sentiment-ed text continuations for IMDB (Maas et al., 2011) movie reviews prompts. We use a sentiment classifier (Sanh et al., 2019) as our reward function that is trained on review texts and sentiment labels from the dataset, which then provides sentiment scores indicating how positive a given piece of text is. For training supervised SFT baselines, we consider only the examples with positive labels. We chose GPT2 (Radford et al., 2019) as the base language model (LM) for this task. We evaluate all algorithms on three metrics: sentiment reward score, perplexity, and output-perplexity.

Next, we consider CommonGen (Lin et al., 2020b), a challenging constrained, text generation task that tests the ability of generative common sense reasoning. We optimize the SPIDER (Liu et al., 2017) reward function, a weighted combination of the CIDEr-D and SPICE metric. We chose T5-base (Raffel et al., 2020) as our base LLM and prefixed each concept set input with: "generate a sentence with:". We evaluate on four metrics: BLEU (Papineni et al., 2002), CIDEr-D (Vedantam et al., 2015), SPICE (Anderson et al., 2016), and BERTScore (Zhang et al., 2019a).

The final task we consider is Reddit TL;DR summarization dataset (Völske et al., 2017) where the objective is to generated summaries. We use the filtered dataset with additional human preference data used in Stiennon et al. (2020). The base LLM that we use for this task is GPT-J (Wang & Komatsuzaki, 2021) and we train all models using LoRA adapters(Hu et al., 2021). We evaluate all algorithms on 5 metrics: reward score, perplexity, output-perplexity, win rate and Rouge (Lin, 2004). For win rate, we use the open source Llama2-13B-chat (Touvron et al., 2023) model as our evaluator model. We compare all algorithm generations to the preferred summary references. Refer to Appendix C.2, for the exact Win Rate prompt, example evaluations and implementation details.

## 6.1 EXPERIMENTAL RESULTS

**RLGF vs. RLHF Performance**  Table 1 and Table 2 compares all of the RLGF algorithms proposed in Section 4 against standard RLHF algorithms and baselines. For all tasks, our $\pi^g$ is SFT which is sub-optimal, performing worse than all RL based algorithms across most lexical and semantic metrics. Utilizing this $\pi^g$, for IMDB, SFT+D$^2$LOLS and PPO$^{++}$ outperform PPO, and for CommonGen, D$^2$LOLS outperforms PPO . Finally, for TL;DR summarization we see that PPO$^{++}$ performs better than PPO as well as a competitive baseline, Best-of-N (Dubois et al., 2023).

| Algorithms | TL;DR Summarization _Semantic and Fluency Metrics_ | | | | | | |
|---|---|---|---|---|---|---|---|
| | RM Score ($\uparrow$) | Perplexity ($\downarrow$) | Output-Perplexity ($\downarrow$) | Win Rate ($\uparrow$) | Rouge 1 ($\uparrow$) | Rouge 2 ($\uparrow$) | RougeL ($\uparrow$) |
| Zero-Shot | 1.57 | 14.07 | 11.51 | 44.12% | 0.27 | 0.07 | 0.18 |
| SFT | 5.68 | 14.09 | 12.81 | 44.29% | 0.34 | 0.25 | 0.25 |
| Best-of-N($N=8$) | 5.98 | 14.09 | 12.86 | 47.60% | 0.36 | 0.13 | 0.27 |
| SFT+PPO | 6.01 | 15.05 | 17.67 | 54.25% | 0.35 | 0.13 | 0.27 |
| SFT+PPO$^{++}$ | 6.11 | 14.53 | 16.15 | 55.01% | 0.36 | 0.14 | 0.27 |
| SFT+AggreVaTeD | 5.93 | 14.69 | 16.41 | 48.98% | 0.36 | 0.15 | 0.29 |

Table 2: **TL;DR Summarization Results:** We report the mean over 1 seed. Our RM Score is under our trained preference reward model and the Win Rate is evaluated by Llama2-13B-Chat. We use SFT+nucleus as $\pi^g$.

Supporting our justification from Section 5, AggreVaTeD improves beyond our guide policy, providing an alternative as a warm-starting methodology to warm-starting with SFT. As shown by Table 7, we see that warm-starting with AggreVaTeD leads to higher performance on IMDB than warm-starting with SFT, a popular learning strategy when performing RL for language (Stiennon et al., 2020; Ouyang et al., 2022). PPO$^{++}$, on the other hand, is better than or competitive to our RL baseline demonstrating a simple, yet powerful alternative to PPO as the RL procedure. Even in practice, we observe the benefit of restarting from an initial state distribution that better covers an optimal policy's state distribution. The combination of these two, D$^2$LOLS, achieves the best of both worlds and fully leverages the capabilities of utilizing a guide policy.

**Reward Optimization Tradeoff** In Figure 2 we evaluate how well RLGF algorithms trade-off optimizing the reward while minimizing the perplexity and kl-constraint $\sqrt{KL}$. For both plots, the top right corner indicates the policy has both high reward and low perplexity and low divergence from $\pi_0$. For each algorithm we plot 5 checkpoints ranging from 20 to 100 iterations. PPO$^{++}$ mostly matches or has higher reward than PPO while maintaining a lower perplexity. Separately, AggreVaTeD trade-offs reward for perplexity, and has comparable reward scores as PPO while drastically reducing its perplexity. For the kl-constraints plot on the left of Figure 2 we see that although PPO has a set of points with high reward, most of these points also have high KL divergences. Whereas, a subset of PPO$^{++}$ matches or has higher reward than PPO while having a lower kl-constraint.

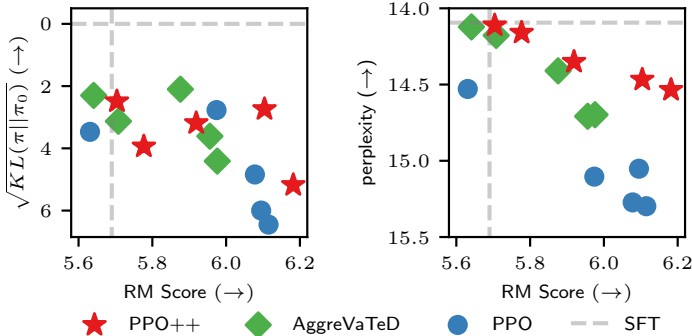

Figure 2: We investigate the reward optimization, kl-constraint, and fluency trade-off in our TL;DR summarization task. The dashed line represents our SFT policy's performance across each metric. Both PPO$^{++}$ and AggreVaTeD learn a policy that has a better trade-off than PPO.

**RLGF Performance on Difficult Prompts** Our evaluation was carried out on the CommonGen task where we categorized the prompts based on their difficulty level. For CommonGen, we classify the prompts into _easy_ and _hard_ based on the number of unseen concepts in the prompt. Specifically, we categorized prompts with 3 concepts as easy and more than 3 concepts as hard. Figure 3 presents a comparison of scores for different algorithms grouped by prompt difficulty. The results reveal a notable performance gap between easy and hard prompts for algorithms such as SFT and PPO,

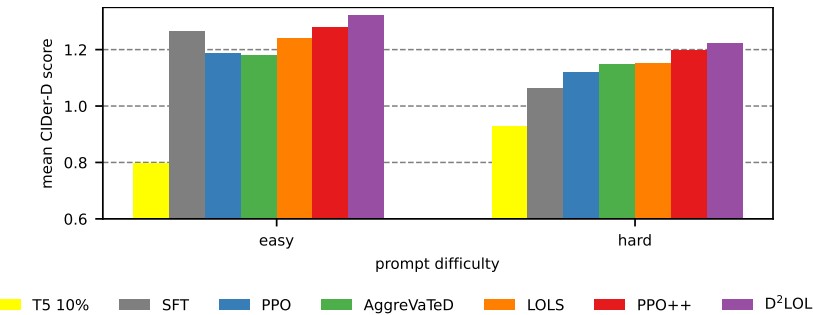

Figure 3: Comparison of CIDer-D scores grouped by prompt difficulty on CommonGen. The performance gap between easy and hard prompts is evident for `SFT`, and `PPO`$^{++}$, while our proposed algorithms `AggreVaTeD`, `LOLS` and `D`$^2$`LOLS` exhibit a significantly smaller gap, showcasing their effectiveness on challenging prompts.

whereas our proposed algorithms `PPO`$^{++}$, `AggreVaTeD`, `LOLS` and `D`$^2$`LOLS` exhibit a significantly smaller gap, with `D`$^2$`LOLS` having the least gap. In other words, even on challenging prompts, our interactive algorithms produce better text continuations. See Appendix E for example generations.

**MLE and KL coefficient Sensitivity**    We test the sensitivity of PPO and RLGF algorithms to two regularization hyperparameters in the $J_{ppo}$ objective, namely the KL coefficient, $\lambda$, and the MLE coefficient, $\eta$. The left 2 plots in Figure 4 show the reward and perplexity when we keep $\eta$ fixed and vary $\lambda$ while the right 2 show the performance when we keep $\lambda$ fixed and vary $\eta$. All RL algorithms are robust to varying KL coefficients. We observe much more instability when relaxing our MLE regularization with PPO and RLGF algorithm's perplexities blowing up.

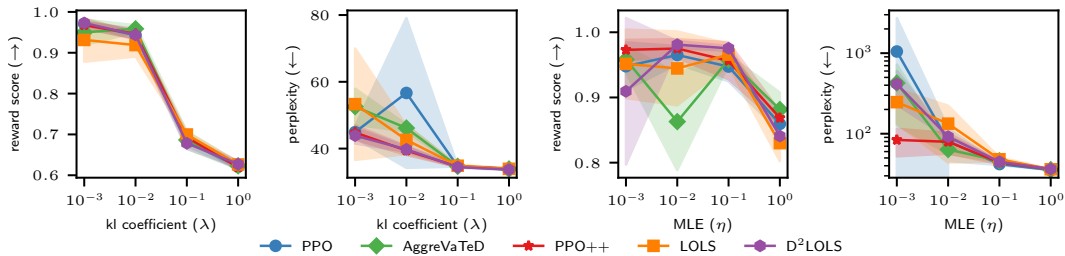

Figure 4: $J_{ppo}$ KL coefficient ($\lambda$) and MLE coefficient ($\eta$) ablation. We show the sensitivity of PPO and RLGF algorithms to each regularization term in the objective. Note that all RL algorithms are robust to changes in KL coefficient with relatively minor changes in the Perplexity while being more sensitive to changes in MLE objective (Right) with blowups in the perplexity.

## 7    CONCLUSION AND FUTURE WORK

We presented a unifying framework of incorporating a guide policy to enhance reinforcement learning for natural language generation. Through theoretical justification and experimental validation, we demonstrate that our RLGF framework can outperform PPO for fine-tuning LLMs. Our proposed algorithms `PPO`$^{++}$ and `D`$^2$`LOLS` only require black-box access to the guide policy and are conceptually simple and easy to implement based on `PPO`. While in our experiment, we demonstrate that supervised fine-tuned models with standard decoding strategies is a good candidate of the guide policy, our framework is general enough to leverage any large LLMs as the guide policy, including those that are not open-sourced. Finally, RLGF's contributions to the broader large language model literature is complementary to model enhancements, dataset improvements, and prompting discoveries such as in-context prompting. We leave it to exciting future work to test the full capabilities of bootstrapping the state-of-the-art advancements in each research direction with RLGF to improve reinforcement learning for natural language generation.

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
