# A ADDITIONAL RELATED WORK

**LLM Alignment** Using RLHF is one idea of aligning LLM with human preferences. The RLHF objective incorporates a KL constraint and is equivalent to minimizing the reverse KL between KL-control distribution and the learner. Minimizing some divergence between policy used for the KL-control and learner policy has been proposed for LLM alignment. (Korbak et al., 2022; Khalifa et al., 2020; Go et al., 2023) propose alignment ideas the attempt to minimize various divergence inspired from maximize entropy RL (Haarnoja et al., 2017; 2018) and Distributional Policy Gradient (DPG) (Barth-Maron et al., 2018). Depending on the chosen divergence, the desired policy behavior may be easy or hard to obtain. Another collection ideas for alignment focus on aspects of the supervised learning data, for example currating the collected data (Zhou et al., 2023; Chung et al., 2022).

**Restart Distribution** On-policy RL algorithms are not able to take advantage of past visited states. But incorporating the ability to reset to any arbitrary state allows on-policy methods to create new states from past visited states (Tavakoli et al., 2018). The core of the idea is to use past visited states to modify the initial state distribution. Our work introduces $\texttt{PPO}^{++}$ which is an algorithm that has no prior over past visited states but (Tavakoli et al., 2018) considers incorporating priories to help decide how to prioritize past visited states to incorporate into the initial state distribution. (Agarwal et al., 2020) showed theoretically that the initial state distribution helps with exploration. Modifying the initial state distribution using restart has seen success in Montezuma Revenge Atari 2600 (a hard exploration problem) and Atari 2600 games more broadly(Popov et al., 2017; Salimans & Chen, 2018; Ecoffet et al., 2019; Florensa et al., 2017).

**NLP with Human Feedback** Learning from human feedback has been studied in the past in the context of bandit feedback (Nguyen et al., 2017; Sokolov et al., 2016), pairwise feedback (Scheurer et al., 2023; Chen et al., 2023) and other feedback forms (Kreutzer et al., 2018a; Sumers et al., 2021; Hancock et al., 2018; Wu et al., 2021). RLHF from has been an active area of research employing RL as the main strategy to align LMs with human preferences (Ouyang et al., 2022; Bai et al., 2022a; Bakker et al., 2022; OpenAI, 2023; Nakano et al., 2021; Wu et al., 2021; Stiennon et al., 2020; Ziegler et al., 2019). A remarkable result in this line of work is ChatGPT (OpenAI, 2023). The general process involves learning a preference reward model induced by human preferences and then finetuning with RL using this learned preference model.

**LLM Finetuning from AI Feedback:** Despite being easier to collect than expert data, high-quality human preference data collection is a key bottleneck of scaling RL finetuning for LLMs. A growing body of work enlists the help of LLMs to augment various parts of the RLHF procedure. ConstitutionalAI and RLAIF (Bai et al., 2022b; Lee et al., 2023a) explores using LLMs to generate preference datasets to do reward model training on while (Roit et al., 2023; Yang et al., 2023; Kwon et al., 2023) finds directly generating reward signals from another LLM to be effective. Separate from this literature, we investigate utilizing direct LLM feedback during the generation process, reminiscent of RL algorithms utilizing expert interactive feedback.

**RL for Text Understanding and Generation:** RL has been used to train text generation models for dialogue (Li et al., 2016), text simplification (Zhang & Lapata, 2017), machine translation (Kiegeland & Kreutzer, 2021; Wu et al., 2016; Shen et al., 2015), image captioning (Ren et al., 2017), question generation (Pang & He, 2021). RL has also been used to create models that take actions given a text such as for instruction following (Hermann et al., 2017; Misra et al., 2017), text games (Narasimhan et al., 2015; Côté et al., 2019; Ammanabrolu & Riedl, 2018), and code generation (Zhong et al., 2017). These methods typically use policy gradient based RL. Recently, (Ramamurthy et al., 2022) studied online RL for text generation across a wide range of tasks, specifically studying Proximal Policy Optimization (PPO) (Schulman et al., 2017). Although the results comparing RL and SL are mixed, we build upon their work and show the benefit of RL and ultimately RLGF outperforming SL and RL. Separately, (Snell et al., 2022) studies offline RL in the context of text generation whereas our work studies the online case.

## B  ADDITIONAL ALGORITHMS

A detailed algorithm for LOLS showing how to combine reinforcement learning and imitation learning differently than $D^2$LOLS. Rather than setting $\alpha$ to be the stopping time to switch from AggreVaTeD to PPO$^{++}$, we have a mixing probability of combining AggreVaTeD and PPO$^{++}$ at every iteration, $\alpha A^{\pi_\theta^t} + (1-\alpha) A^{\pi^g}(s,a)$. As discussed in Section 5, we find that LOLS underperforms $D^2$LOLS, even in practice.

---

**Algorithm 4** LOLS: combine PPO and AggreVaTeD

---

1: **Input:** $\pi_\theta$, reference $\pi^g$, iterations T, dataset $\mathcal{D} = \left\{(x^i, y^i)\right\}_{i=1}^{N}$
2: **Input:** mixing parameter $\beta_1 \in [0,1]$, mixing parameter $\beta_2 \in [0,1]$, mixing prob $\alpha$
3: **for** t = 0,1,...,T-1 **do**
4:
   ▷ PPO$^{++}$
5:     Rollin with $\beta_1\pi^g + (1-\beta_1)\pi_\theta^t$ starting from $x \sim \mathcal{D}$
6:     Rollout with $\pi_\theta^t$ to collect trajectories
7:     Update $V_\phi^{\pi_\theta^t}$ with trajectories and compute advantage estimates $A^{\pi_\theta^t}$
8:
   ▷ AggreVaTeD
9:     Rollin with $\beta_2\pi_\theta^t + (1-\beta_2)\pi^g$ starting from $x \sim \mathcal{D}$
10:    Rollout with $\pi^g$ to collect trajectories
11:    Update $V_\phi^{\pi^g}$ with trajectories and compute advantage estimates $A^{\pi^g}(s,a)$
12:
   ▷ Mix Update
13:    Update $\pi_\theta$ using PPO loss with $\alpha A^{\pi_\theta^t} + (1-\alpha) A^{\pi^g}(s,a)$

---

## C    ADDITIONAL EXPERIMENTAL DETAILS

### C.1    KL REWARD CONSTRAINT

In addition to sequence-level task rewards, per-token KL rewards are applied to prevent the policy $\pi$ from deviating too far from the pre-trained LM $\pi_0$, following the works Ziegler et al. (2019); Ouyang et al. (2022); Ramamurthy et al. (2022). Formally, regularized reward function is defined as: $\hat{R}(s_t, a_t, y) = R(s_t, a_t, y) - \lambda\text{KL}\left(\pi(a_t|s_t)||\pi_0(a_t|s_t)\right)$ where KL $\left(\pi(a_t|s_t)||\pi_0(a_t|s_t)\right) = (\log\pi(a_t|s_t) - \log\pi_0(a_t|s_t))$ and $\lambda$ is the KL coefficient Ouyang et al. (2022). Note we used use a fixed KL coefficient rather than an adaptive controller.

### C.2    TASK DETAILS

| Task | Train/Val/Test | Prompt | Gen. Length |
|---|---|---|---|
| IMDB | 25K/5K/5K | Partial movie review up to 64 tokens | 48 |
| CommonGen | 32651/993/1497 | "Generate a sentence with: " set of 3-5 concepts | 20 |
| TL;DR | 117000/6450/6550 | "TL;DR: " | 50 |
| TL;DR Preference | 92500/3300/8300 | "TL;DR: " | N/A |

Table 3: Train, val, test splits, prompts, and max generation length used for each task.

**IMDB:**    We experiment on the IMDB dataset for positive movie review generation. As shown in Table 3, the dataset consists of 25k training, 5k validation and 5k test prompts of movie review text with either positive or negative sentiment labels. As in put to our models, we use partial movie reviews that are at most 64 tokens long and ask the model to complete the review with a positive sentiment with at most 48 generated tokens.

**CommonGen:**    CommonGen Lin et al. (2020b) is a common sense text generation task where the model is given a set of concepts (i.e. hockey, rink, game) and is asked to generate a semantically correct sentence using those concepts (i.e. the hockey team played a game at the rink). We follow the same splits as the dataset creators and refer the readers to Table 1 of Lin et al. (2020b) for more in-depth statistics of the dataset. In our experiments, we prompted out models with "*generate a sentence with:* " and generated at most 20 tokens. We chose this generation length based on the maximum token length of the references in the training dataset.

**TL;DR Summarization:**    Following Stiennon et al. (2020), we evaluate on the summarization task. We use `CarperAI/openai_summarize_comparisons` for the preference reward training dataset and `CarperAI/openai_summarize_tldr` for the RL training dataset. For the SFT model that we use for our starting policy and our guide policy, we use the publicly available checkpoint `CarperAI/openai_summarize_tldr_sft`. We truncated/padded each prompt to 500 tokens on the GPT-J 6B tokenizer.

We first train our reward model using LoRA adapters. Our reward training is 1 epoch and where we got 70% accuracy on the test set. With this reward model we run all of our experiments where our policy and critic are both LoRA adapters trained on top of SFT checkpoint.

**Win Rate:**    We calculated the win rate against the dataset references using Llama2-13B-chat (Touvron et al., 2023) publically available on HuggingFace. Following DPO (Rafailov et al., 2023), we prompt the model with instructions, 2 summaries (A) and (B), and instructions on how to answer. We randomize which summary is (A) or (B) when calculating the win rate over the test set. Below is our prompt skeleton:

```
<<SYS>>
You are an expert summary evaluator and can consistently
distinguish between good and bad summaries. You provide
informative, correct evaluations.
<<\SYS>>

Task: Judge the quality of two TLDRs, choose the options
among (A) or (B)
context: [context]
tldr (A): [summary 1]
tldr (B): [summary 2]
FIRST provide a one-sentence comparison of the two summaries,
explaining which you prefer and why. SECOND, on a new line,
state only (A) or (B) to indicate your choice. Your
response should use hte format:
Comparison: <one-sentence comparison and explanation>
Preferred: <(A) or (B)>
```

## C.3  IMDB - ALGORITHM DETAILS

Table 4 lists the hyperparameters used in our IMDB experiments. Note that we used the same parameters here for all guide policies. Across all algorithms, we shared the same parameters as the ones we used for our `PPO` baseline. Finally, we use top-k sampling with $K = 50$ as the decoding method and for fair comparison, we keep this setting for all methods.

| Setting | Values |
|---|---|
| model | GPT2 |
| PPO | steps per update: 1280 |
| | total number of steps: 128000 |
| | batch size: 64 |
| | epochs per update: 5 |
| | learning rate: 1e-6 |
| | discount factor: 0.99 |
| | gae lambda: 0.95 |
| | clip ratio: 0.2 |
| | value function coeff: 0.5 |
| | $\lambda$: 0.001 |
| | $\eta$: 0.1 |
| PPO$^{++}$ | Mixing Parameter ($\beta$): 0.2 |
| AggreVaTeD | Mixing Parameter ($\beta$): 0.8 |
| LOLS | Mixing Probability ($\alpha$): 0.8 |
| D$^2$LOLS | Stopping Time Iteration ($\alpha$): 20 |
| decoding | sampling: true |
| | top k: 50 |
| | min length: 48 |
| | max new tokens: 48 |
| tokenizer | padding side: left |
| | truncation side: left |
| | max length: 64 |

Table 4: Hyperparameters used for IMDB. Note that PPO$^{++}$, AggreVaTeD, LOLS, and D$^2$LOLS all share the same PPO parameters. All processes use the same decoding and tokenizer parameters.

## C.4 COMMONGEN - ALGORITHM HYPERPARAMETERS

| Setting | Values |
|---|---|
| model | T5 |
| PPO | steps per update: 663,552 
 total number of steps: 66,355,200 
 batch size: 2048 
 epochs per update: 1 
 learning rate: Linear decay 1e-5 
 discount factor: 0.99 
 gae lambda: 0.95 
 clip ratio: 0.4 
 value function coeff: 3.0 
 $\lambda$: 0.001 
 $\eta$: 0.1 |
| PPO$^{++}$ | Mixing Parameter ($\beta$): 0.2 |
| AggreVaTeD | Mixing Parameter ($\beta$): 0.8 |
| LOLS | Mixing Probability ($\alpha$): 0.8 |
| D$^2$LOLS | Stopping Time Iteration ($\alpha$): 20 |
| decoding | num beams: 5 
 min length: 5 
 max new tokens: 20 |
| tokenizer | padding side: left 
 max length: 20 |

Table 5: Hyperparameters used for CommonGen. Note that PPO$^{++}$, AggreVaTeD, LOLS, and D$^2$LOLS all share the same PPO parameters. All processes use the same decoding and tokenizer parameters.

Table 5 lists the hyperparameters used in our CommonGen experiments. Note that we used the same parameters here for all guide policies. Across all algorithms, we shared the same parameters as the ones we used for our PPO baseline. Finally, we use beam search with the number of beams = 5 as the decoding method for inference. Note that for training, we still used softmax sampling with default temperature. For fair comparison, we keep this setting for all methods. Finally, note that for CommonGen, we set the KL coefficient to 0.

## C.5 TL;DR SUMMARIZATION - ALGORITHM HYPERPARAMETERS

Table 6 lists the hyperparameters used in our TL;DR summarization experiments. Note that we used the same parameters here for all guide policies. Across all algorithms, we shared the same parameters as the ones we used for our PPO baseline. The reward LoRA adapter was used to pretrain our model. During RL training, we kept the reward adapter frozen.

| Setting | Values |
|---|---|
| model | GPT-J |
| PPO | steps per update: 70,400
total number of steps: 7,040,000
batch size: 128
epochs per update: 4
learning rate: 1e-5
discount factor: 1.0
gae lambda: 0.95
clip ratio: 0.2
value function coeff: 0.2
$\lambda$: 0.002
$\eta$: 0.1 |
| PPO$^{++}$ | Mixing Parameter ($\beta$): 0.6 |
| AggreVaTeD | Mixing Parameter ($\beta$): 1.0 |
| decoding | max new tokens: 50 |
| tokenizer | padding side: left
truncation side: right
max length: 50 |
| RM LoRA | r: 32
$\alpha$: 32
dropout: 0.0 |
| Policy LoRA | r: 8
$\alpha$: 64
dropout: 0.1 |
| Critic LoRA | r: 8
$\alpha$: 64
dropout: 0.1 |

Table 6: Hyperparameters used for TL;DR summarization. Note that PPO$^{++}$, AggreVaTeD all share the same PPO parameters.

## D  IMDB QUALITATIVE EXAMPLES

**Sample** 1
**Prompt:** not a complex character, but totally rotten. The relation between him and Gisburne is just hilarious. Actually, just looking at de Rainault sitting in his throne, bored, glaring, makes me laugh even before he has said anything. Another actor that deserves extra praise is Ray Winstone as Will Scarlet. You
**GPT2:** know, when you think he's a bad actor, and when you think the script is all about the bad actors. Scarlet is the typical actor with a weak, pathetic voice, who fails to show emotion or self-respect, and who
**BC:** will not be disappointed. The opening scenes are amazing and the final scene of the film is amazing. Every single minute of this film is a treat with every emotion the characters are experiencing. From the first frame of
**PPO:** know he likes old movies. I gave this film a 3 and a half out of 10. And, of course, I don't blame you for falling asleep! I gave it a 10 because of my favorite
**PPO+:** see he seems only interested in money, which really makes the two seem like friends. One particularly funny scene involves Winstone and the rest of the cast in their studio for the first time since "Rampage". I absolutely LOVE the way
**AggreVaTeD:** will laugh heartily; he is brilliant and always entertaining. I also think the film is just wonderful. I love what Van Dyke is able to bring to the film. If you want to go great, I'm sure Van Dyke
**LOLS:** great film. I absolutely loved his style and direction. One of my favorite movies of all time. The first 25 minutes are really good and really enjoyable. It's actually very touching and engaging. The cast, including James Gandolfini
**D2LOLS:** could never have come across this again. It was one of the funniest films I've ever seen, and it remains my favorite movie ever. I am absolutely honored and very excited to see it again!!! I really love it and look forward

**Sample** 2
**Prompt:** comes from Zekeria Ebrahimi, who plays the young Amir. It's not that it's bad, it's just that it's not as effective as it could be. He just can't translate the guilt that consumes Amir to the audience. In fact, I think I might have been a little lost at
**GPT2:** first. The movie has too much going for it, too many unanswered questions and some of the characters not getting the emotional payoff in a reasonable enough manner. It's nice to see a filmmaker who isn't a hater, but I feel
**BC:** the second half as it wasn't shown at all until the very end, at best. I liked the plot ( which I think was very well done, considering it's a "bad cop story"). It's not
**PPO:** first. After listening to the great Ghazala, who's brilliant in her acting, I am very impressed with her acting performance. I loved how you have a young and very attractive girl named Zekeria who is portrayed quite well and
**PPO+:** last week's deadline. It's quite a clever and good film, as well as an entertaining one. 4/10. Recommended. 8/10
**AggreVaTeD:** first. The movie really deserves the credit – it really is a very moving story and a really moving movie that is an amazing film. I personally love Amir and think it's an extremely powerful film. It's one of the things that makes
**LOLS:** some points. As a whole it's pretty enjoyable and is very well-acted and well edited, and is quite a compelling, quirky and fun film. And just as important, I found some very well-acted, and well-acted
**D2LOLS:** first, when Amir realizes what's really going on. There are many terrific performances in this beautifully directed film. I can't say which is one of the best, which is certainly a shame because it's an enjoyable

**Sample** 3
**Prompt:** Well, now I finally HAVE seen the true garbage version – and it's Paul Schrader's DOMINION: PREQUEL TO THE EXORCIST. It was relentlessly talky, uninteresting, and insipid. Stellan Skarsgard's troubled priest was nowhere near as
**GPT2:** interesting as the other characters in his book – and most of the scenes in the book were just so poorly edited and over-the-top; the acting was too, horribly bad, and most of the dialogue was thrown out in a couple
**BC:** well developed as the novel and story had – but did bring some of the best bits to the screen. I watched the movie expecting a story – "The God" – but found it just an excuse for schlock
**PPO:** good (I guess), if anything, the sort of low-brow, uninteresting sort of horror that made an original film "The Cat With the Feeling" look like some sort of cheesy 80's horror flick from 1999.
**PPO+:** exciting. And in addition, the only really really interesting character in this movie was Paul Schrader, who is at his worst as a creepy, self-absorbed man in a bar. I can't say I've ever liked a
**AggreVaTeD:** original or entertaining as I'd anticipated, and in the end, I got to see some interesting and engaging dialog. I loved that scene with Schrader explaining how it would be fun for him to make a movie that would make people think twice
**LOLS:** good as his master. Still, the film's message is very much worth exploring. It was the perfect love story for me, as you would expect. I'm an evangelical and am an excellent fan of all things Christian. I love reading
**D2LOLS:** brilliant as the others, as well as beautifully acted. Of course, some points in the film are so well acted and acted that, despite all its flaws, this is a very touching film. The cinematography is

Figure 5: **Qualitiative Analysis of IMDB**

# E    COMMONGEN QUALITATIVE EXAMPLES

**Sample** 1
**Prompt:** generate a sentence with: dress perform stage.
**T5:** dress up and perform on stage
**BC:** A young girl in a black dress is performing on a stage.
**PPO:** performers dress up on the stage.
**PPO**+: a woman performing on the stage dressed up.
**AggreVaTeD:** young woman performing on stage in a black dress.
**LOLS:** people performing on the stage in a dressed up.
**D2LOLS:** A woman is performing on the stage in a dress.

**Sample** 2
**Prompt:** generate a sentence with: ball fall throw.
**T5:** ball falls off the ground
**BC:** A man throws a ball that has fallen to the ground.
**PPO:** the ball falling from the throw.
**PPO**+: the ball is thrown and the ball falls.
**AggreVaTeD:** A man is throwing the ball into the ground.
**LOLS:** throwing the ball after it fell from the sky.
**D2LOLS:** A man is throwing the ball after it fell.

**Sample** 3
**Prompt:** generate a sentence with: arm chest fold.
**T5:** arm folds in the chest
**BC:** He folds his arms over his chest, then he folds his arms over.
**PPO:** folded the arms in the chest.
**PPO**+: a man with his arms folded in the chest.
**AggreVaTeD:** folding his arm over his chest.
**LOLS:** A man folds his arms in the chest.
**D2LOLS:** A man with his arms folded in the chest.

Figure 6: **Qualitiative Analysis of CommonGen**

# F    ADDITIONAL RESULTS

We further go on to compare starting from either a `SFT` warmstarted policy of `AggreVaTeD`. We show that for IMDB, we obtain better performance when performing RL after `AggreVaTeD` rather than `SFT`.

| Alg | Semantic Score |
|---|---|
| `SFT`+`PPO` | $0.767 \pm 0.018$ |
| `AggreVaTeD` + `PPO` | $0.863 \pm 0.007$ |
| `SFT`+`PPO`$^{++}$ | $0.883 \pm 0.011$ |
| `D`$^2$`LOLS`(i.e. `AggreVaTeD` + `PPO`$^\dagger$) | $\mathbf{0.896 \pm 0.012}$ |

Table 7: Warmstarting with `SFT` or `AggreVaTeD`: Results of running `PPO` or `PPO`$^{++}$ after warmstarting with either `SFT` or `AggreVaTeD`. For both `PPO` and `PPO`$^{++}$, warmstarting with `AggreVaTeD` yields the better results.