# OpenReview forum: "Learning to Generate Better than your Large Language Models"
_ICLR.cc/2024/Conference — Submitted to ICLR 2024_

### Official Review · Reviewer_eNmj · 2023-10-28

**Soundness:** 2 fair
**Presentation:** 3 good
**Contribution:** 2 fair
**Rating:** 3
**Confidence:** 2

**Summary:**

This paper proposes D^2LOLS, an RL optimization technique that can take advantage of the use of a guide policy to overcome some of the limitations of RL algorithms currently used with LLMs, such as PPO. The basic intuition is to use guidance from a superior model to ensure that the target model does better than the guidance model, by biasing rollouts to more optimal paths that the superior model might have taken and updating the target model with the reward within this superior set of rollouts. Related work is reviewed, and the proposed method is described clearly in steps as updates and combinations of previous work. Theoretical justification is given for why D^2LOLS should do better, due to restarting on states according to the guidance model, though an assumption is made that does not seem to be clearly evidenced, see Questions.

Experiments are performed on 3 datasets: IMDB, CommonGen, and TL;DR. The first two are rather simple (write a movie review with a given sentiment, include words in a given generated sentence), and IMDB is further handicapped to only consider one sentiment. Furthermore, the gains over baselines are relatively small. Additionally, the authors do a study of the tradeoff between optimizing for the RL reward and perplexity, which can be interpreted as a partial proxy for the acceptable space of generated outputs. In a similar vein, hyperparameter sensitivity is discussed with further experiments.

The authors conclude by suggesting that the proposed technique allows for superior optimization using only black box access to a guidance policy, making it especially useful in the era of powerful LLMs which can only be accessed through APIs.

**Strengths:**

- Clear explanation of both the algorithm and the key intuitions

- The basic idea is definitely worth pursuing—using a guide for rolling out possible scenarios in order to ensure some coverage of the space of near-optimal policies is clearly something that might benefit current RL methods in LLMs.

- The study of reward optimization tradeoff is interesting and very welcome. It’s pretty clear that our metrics only work under “normal conditions” where complete gibberish isn’t being measured, so using the KL divergence as the other axis shows a potential pareto frontier that we’re working with in these problems.

- The fact that the proposed technique seems to generalize slightly better than previous techniques to harder examples (with less train-test overlap) on CommonGen is interesting, and does validate the technique to some extent, though the margin is relatively small.

**Weaknesses:**

- The benchmarks used are somewhat lacking. For text generation in 2023, most comparisons for state of the art methods are made by comparisons to other LLMs because fixed metrics have been found to be lacking. A comparison using a framework like Chatbot Arena (Zheng et al. 2023). would have made the results significantly more convincing. These often use GPT-4 as an evaluator, which has now been shown to be provisionally better than traditional metrics (Liu et al 2023, Min et al 2023, Zhou et al 2023, inter alia).

- Even with these quite simple benchmarks, the resulting differences on metrics are small for the task where control is more significant, CommonGen. In IMDB the task-specific metric really only checks if the sentiment score is correct, which is very little information to be using to validate outputs.

- The fact that perplexity goes up but output perplexity goes down on IMDB is worrying. I can understand why perplexity would go up: the model is likely collapsing onto a smaller subspace of possibilities, which may still be higher quality. However, the fact that the output perplexity also goes down is worrying: it indicates that after optimization the resultant model is more predictable to LMs which is usually not a good sign for tasks that LMs are bad at in the first place. The results on TL;DR are similarly quite small.

- I’m disappointed to see there’s no study of how this technique scales across model sizes: one could imagine doing this for smaller models, incurring relatively low compute expenses, but no such experiments were conducted. As it is, it is not clear how much these results will generalize to the ever-growing zoo of LLMs.

Works Referenced

Liu, Yang, Dan Iter, Yichong Xu, Shuohang Wang, Ruochen Xu, and Chenguang Zhu. 2023. “G-Eval: NLG Evaluation Using GPT-4 with Better Human Alignment.” arXiv [cs.CL]. arXiv. http://arxiv.org/abs/2303.16634.

Min, Sewon, Kalpesh Krishna, Xinxi Lyu, Mike Lewis, Wen-Tau Yih, Pang Wei Koh, Mohit Iyyer, Luke Zettlemoyer, and Hannaneh Hajishirzi. 2023. “FActScore: Fine-Grained Atomic Evaluation of Factual Precision in Long Form Text Generation.” arXiv [cs.CL]. arXiv. http://arxiv.org/abs/2305.14251.

Zheng, Lianmin, Ying Sheng, Wei-Lin Chiang, Hao Zhang, Joseph E. Gonzalez, Ion Stoica. “Chatbot Arena: Benchmarking LLMs in the Wild with Elo Ratings.” 2023. https://lmsys.org/blog/2023-05-03-arena/.

Zhou, Xuhui, Hao Zhu, Leena Mathur, Ruohong Zhang, Haofei Yu, Zhengyang Qi, Louis-Philippe Morency, et al. 2023. “SOTOPIA: Interactive Evaluation for Social Intelligence in Language Agents.” arXiv [cs.AI]. arXiv. http://arxiv.org/abs/2310.11667.

**Questions:**

- On page 6 you write “While we do not expect the SFT policy π g is as good as the optimal π ⋆ , it is reasonable to expect that d π g provides coverage to d π ⋆ .” Why is this reasonable to expect? It is not a prior clear to me that language models cover the space of optimal solutions well, rather than only covering a small percentage of them. Where is the evidence for this claim?

-  What model do you use to calculate output perplexity? I only see you mention GPT-J in passing as an example, not definitively stating what model was used.

-  On page 7 you write “For training supervised SFT baselines, we consider only the examples with positive labels.”—This is a somewhat unorthodox decision, as it gives little for the model to contrast with or for the user to control at inference time. It basically reduces this to a finetuning problem. Why was this choice made?

---

> ### Author Response · Authors · 2023-11-19
> **Response Part 1**
>
> Thank you for your review. We hope to address your concerns here
>
> __Chatbot Arena with GPT4 Evaluation__
>
> Thank you for your suggestions. We would like to clarify the scope of our paper:  we focuses on RL finetuning of specific downstream tasks (e.g., generating positive reviews or reddit post summary) instead of generating a general purpose chat model. None of our tasks were for dialogue and general chat/instruction following and thus we don’t find Chatbot Arena to be a relevant evaluation platform for this work. Our evaluation and tasks closely follow other published RLHF papers in 2023, e.g., RL4LM [1],  which is published in ICLR 2023.
>
> Also we aimed to use open-source models for both training and evaluation of our models. Open-source models remain publicly available, guaranteeing the reproducibility of our experiments and results. Whereas closed source models access can be depreciated [2] at any time, making it impossible to reproduce our results.
>
> [1] Is Reinforcement Learning (Not) for Natural Language Processing: Benchmarks, Baselines, and Building Blocks for Natural Language Policy Optimization, Ramamurthy et al. 2023
>
> [2] https://openai.com/blog/gpt-4-api-general-availability
>
> __Performance Gain__
>
> Please see our updated results we posted along with the rebuttal for more results with TL;DR. For a quick summary, we show that when doing best-of-n sampling with our RL policies (PPO included), PPO++ shows greater performance improvement than PPO.
>
> __Perplexity Goes up but Output Perplexity does down__
>
> We would like to clarify that perplexity is a measure of fluency rather than a measure of how well a generation does on a task. Our evaluation model for output perplexity is GPT2-large. While we agree GPT2-large is not good at the task in terms of optimizing the reward, it is still a valid model that has been widely used to evaluate whether or not the generation is natural and fluent (e.g., see [1-10]).
>
> The overall objective that we are optimizing for is $J_{ppo}$, which includes three terms: reward, KL-constraint (i.e., a fluency constraint), and MLE-constraint (i.e., a fluency constraint). We attempted to capture each term’s significance during evaluation through reward, output perplexity, and perplexity respectively. Both output perplexity and perplexity are not perfectly aligned with the terms in the objective but are somewhat correlated. The policy that we are computing a KL-constraint with respect to is different from the output perplexity policy. Furthermore, we would like to point out that there could be high-scoring generations with low fluency as well as high-scoring generations with high fluency. The output perplexity being better after the RL procedure shows that we are maintaining fluent generations while improving the reward that we are optimizing on.
>
> [1] DExperts: Decoding-Time Controlled Text Generation with Experts and Anti-Experts by Liu et al. 2021
>
> [2] PLUG AND PLAY LANGUAGE MODELS: A SIMPLE APPROACH TO CONTROLLED TEXT GENERATION by Dathathri et al. 202
>
> [3] Plug-and-Blend: A Framework for Plug-and-play Controllable Story Generation with Sketches by Lin et al. 2021
>
> [4] POINTER: Constrained Progressive Text Generation via Insertion-based Generative Pre-training by Zhang et al. 2020
>
> [5] COCON: A SELF-SUPERVISED APPROACH FOR CONTROLLED TEXT GENERATION by Chan et al 2021
>
> [6] FUDGE: Controlled Text Generation With Future Discriminators by Yang et al. 2021
>
> [7] Mix and Match: Learning-free Controllable Text Generation using Energy Language Models by Mireshghallah et al. 2022
>
> [8] A DISTRIBUTIONAL APPROACH TO CONTROLLED TEXT GENERATION Khalifa et al. 2021
>
> [9] On Learning Text Style Transfer with Direct Rewards by Liu et al 2021
>
> [10] MASKGAN: BETTER TEXT GENERATION VIA FILLING IN THE ___ Fedus et al. 2018
>
> __Scaling Law Testing__
>
> Here are the LLM breakdowns for our experiments
> - IMDB: GPT2-base (117M parameters, Causal model)
> - CommonGen: T5-base (220M parameters, Seq2Seq model)
> - TL;DR: GPTJ (6B parameters, Causal model)
>
> We have the exact hugging-face model specifications in the implementation details in the appendix. Note that we aimed to try varying types of LLMs and sizes of LLMs across our experimentation. We agree that it would be interesting to see how the performance scales w.r.t. Policy model size, but in this work we hoped to focus on improvements beyond PPO when all design choices were held equal.
>
> __Output Perplexity__
>
> We computed Output Perplexity with GPT2-Large. We thank the reviewer for pointing this out and will make this clearer in the final version.

---

> ### Author Response · Authors · 2023-11-19
> **Response Part 2**
>
> **$\pi_{SFT}$  reasonably covers  $\pi^{*}$**
>
> Here we assume $\pi^*$ as the policy that generates the references in the dataset. Then under the assumption that $\pi_{SFT}$ is well trained on the task with references as labels, we claim that $\pi_{SFT}$ has non-zero support over the space of generations that $\pi^*$ may cover. This claim is empirically supported by the fact that SFT has bounded non-infinity validation perplexity: given the definition of perplexity, the only way we can have bounded validation perplexity is for our model to have non-zero probability of generating the references. In other words, this means that the SFT model does have non-zero probabilities generating the references in the dataset.
>
> __Training on Positive__
>
> We would like to clarify that this choice is standard for the IMDB positive text generation task. The original IMDB dataset does not have any reviews that start negative and finish positive. Thus when training an SFT model, it is standard to focus on the positive -> positive reviews as the task-relevant data for supervised learning. This was done in RL4LMs [1], Direct Preference Optimization [2], as well as popular RLHF codebases such as TRLx [3].
>
> [1] Is Reinforcement Learning (Not) for Natural Language Processing: Benchmarks, Baselines, and Building Blocks for Natural Language Policy Optimization, Ramamurthy et al. 2023
>
> [2] Direct Preference Optimization: Your Language Model is Secretly a Reward Model, Rafailov et al. 2023
>
> [3] https://github.com/CarperAI/trlx

---

> ### Comment · Area_Chair_h2oy · 2023-11-20
>
> Hello, reviewer. Please review the author's response. Does the response address your concern about the experimental results?

---

> ### Comment · Reviewer_eNmj · 2023-11-20
>
> This response, while clarifying some points, does not address my concerns. Let's consider each in turn.
>
> > - The benchmarks used are somewhat lacking. For text generation in 2023, most comparisons for state of the art methods are made by comparisons to other LLMs because fixed metrics have been found to be lacking. A comparison using a framework like Chatbot Arena (Zheng et al. 2023). would have made the results significantly more convincing. These often use GPT-4 as an evaluator, which has now been shown to be provisionally better than traditional metrics (Liu et al 2023, Min et al 2023, Zhou et al 2023, inter alia).
>
> While the authors immediately mention that ChatBot Arena is for dialogue (fair enough, but I did not ask them to use ChatBot Arena, just a similar setup), the reality is there is still no direct qualitative study. Authors point out that they tried to use open-source models. Fair enough, but then human evaluations for a qualitative study of what these model outputs actually look like should be done. The scores given on these limited set of tasks are simply insufficient to conclude much.
>
> > - Even with these quite simple benchmarks, the resulting differences on metrics are small for the task where control is more significant, CommonGen. In IMDB the task-specific metric really only checks if the sentiment score is correct, which is very little information to be using to validate outputs.
>
> Even with the extra results, the gains are small.
>
> > - The fact that perplexity goes up but output perplexity goes down on IMDB is worrying. I can understand why perplexity would go up: the model is likely collapsing onto a smaller subspace of possibilities, which may still be higher quality. However, the fact that the output perplexity also goes down is worrying: it indicates that after optimization the resultant model is more predictable to LMs which is usually not a good sign for tasks that LMs are bad at in the first place. The results on TL;DR are similarly quite small.
>
> In their response, the authors describe how perplexity is a measurement of fluency (which is not exactly true, but it can be a decent proxy), and how this is not a problem. I disagree—we have seen almost no evidence that the resulting generations are actually desirable. Again, the lack of a qualitative comparison of what generations actually look like is the only way to find out whether models are over-optimizing certain metrics, which is precisely what the objective incentivizes. In their response, the authors justify the metrics by saying they are close to the training objectives. This is not the point—the question is whether the result of the objective is even desirable in the first place, for which there is still little evidence.
>
> > - I’m disappointed to see there’s no study of how this technique scales across model sizes: one could imagine doing this for smaller models, incurring relatively low compute expenses, but no such experiments were conducted. As it is, it is not clear how much these results will generalize to the ever-growing zoo of LLMs.
>
> In response to my concerns about scale, the authors listed the specifications of the models used, suggesting that these details were in the appendix. I am aware of the models used, but using different models of slightly different sizes on different datasets does not tell us whether scale will impact the use of these techniques. Indeed it actually makes it impossible to deconflate whether the specific model being used is the cause of any gains achieved, rather than the technique itself.
>
>
> My score remains unchanged.

---

> > ### Author Response · Authors · 2023-11-21
> >
> > Thank you for the reviewer's prompt and thorough responses.
> >
> > __Evaluation__
> >
> > Thank you for clarifying. We agree with the reviewer that reproducing/supporting our quantitative results with strong qualitative evaluations such as human evaluation would strengthen the results in this paper. If not Chatbot Arena, could the reviewer provide examples of other frameworks we could use for tasks?
> >
> > We would like to clarify that both IMDB and CommonGen are part of the GRUE benchmark suite [1] for text generation that was also published in 2023. Furthermore, in [1], the original work evaluated if the automated metrics correlated with human judgments within the benchmark and found that they did. TL;DR summarization is still a relevant task to evaluate RLHF algorithms [2,3].
> >
> > [1] Is Reinforcement Learning (Not) for Natural Language Processing: Benchmarks, Baselines, and Building Blocks for Natural Language Policy Optimization, Ramamurthy et al. 2023
> >
> > [2] Statistical Rejection Sampling Improves Preference Optimization, Liu et al. 2023
> >
> >
> > [3] Direct Preference Optimization: Your Language Model is Secretly a Reward Model, Rafailov et al. 2023
> >
> > __Perplexity__
> >
> > Thank you for clarifying the point of the question. We agree that perplexity is a good proxy to measure fluency and the reviewer is satisfied with the quantitative results of the paper. For measuring how desirable this work's generations are given, similar to the evaluation response, we agree that human evaluation would strengthen the results. Currently, we have example generations from each algorithm from the test set of each task. We will add more in the final version and release generations with our open-sourced models.
> >
> > __Investigating Scaling Laws__
> >
> > We agree with the reviewer that investigating scaling laws would be valuable and are excited about this future work. We wish to emphasize that in this work we focused on improvements beyond PPO when all design choices were held equal. The scales for our models followed previous works in the literature including using exact models proposed and released by the GRUE benchmark suite [1]. We still believe that improvements beyond PPO shown at the same model scale for a given task is orthogonal to the investigation of how RL/RLGF algorithms scale with model size.
> >
> > [1] Is Reinforcement Learning (Not) for Natural Language Processing: Benchmarks, Baselines, and Building Blocks for Natural Language Policy Optimization, Ramamurthy et al. 2023

---

### Official Review · Reviewer_7XES · 2023-10-29

**Soundness:** 2 fair
**Presentation:** 2 fair
**Contribution:** 2 fair
**Rating:** 5
**Confidence:** 3

**Summary:**

The paper proposes an RL framework for natural language generation, which involves two distinct policies: one providing a trajectory from a given prompt and the other completing the sequence from a state sampled from the trajectory. In this framework, one policy is trained, while the other remains fixed and serves as a guiding policy, producing useful states for the learning process of the other policy. The authors introduce three variants within the framework and demonstrate their effectiveness on three distinct NLP tasks.

**Strengths:**

- The idea is simple and interesting. It could easily lead to others building on the core concept and approach.
- The paper reviews a line of literature on imitation learning and presents a connection between those and the proposed framework.

**Weaknesses:**

- The motivation is not clear.  Why guide policy should be integrated into RL finetuning especially in text generation? How the rollin & rollout scheme leads better text generation than PPO?
- The theoretical justification section of the paper does not self-contained enough for readers.
- Performance gain seems marginal compared to PPO.
- Lack of ablation study for the mixing parameter $\beta$ which might be crucial in the framework.

**Questions:**

- It would be interesting to observe the utilization of a more powerful guide policy,  such as larger LMs like LLaMA, than the other policy in the framework.
- I am confusing on the setting $\beta = 1.0$ of AggreVaTeD in TL;DR. How $\pi_{\theta}$ of AggreVaTeD can be trained in this setting despite that it does not use $\pi_{\theta}$ at all?
- missing result for SFT + D2LOLS in TL;DR
- missing related work that presents similar concept of the proposed algorithm
    - Selective Token Generation for Few-shot Natural Language Generation
- typo
    - section 6.1: kl-constriant → kl-constraint

---

> ### Author Response · Authors · 2023-11-19
>
> Thank you for your review. We hope to address your concerns here
>
> __Motivation__
>
> The main motivation of this work is to improve on-policy reinforcement learning algorithms that have contributed to the success of ChatGPT. Despite their success, on-policy reinforcement learning algorithms can be very inefficient in exploration, and our goal is to address this issue. If we can address this issue then we can train policies that perform better. Our key observation, stated in our introduction, is that modern LLMs have impressive general language capabilities, which can help guide exploration for training models with reinforcement learning. Because there are numerous ways to provide guidance, we evaluate all combinations of guide paradigms (i.e., rollin-rollout interaction paradigms) to see which ones are better suited for text-generation.
>
> The rollin-rollout scheme offers a paradigm to investigate various different possible ways that a guide policy can help with exploration. When rollin and rollout are with the learner policy then we reduce to normal PPO without any guidance. Rollin with a guide policy and rollout with a learner policy, means the guide policy provides more referent states where the learner policy can reset and perform optimization. These good reset states equate to the learner policy receiving high rewards much early during training. Seminal RL work like CPI [1] has shown that a good reset distribution can make RL to learn a higher quality policy. On the other hand, rollin with the learn policy and rollout with the guide policy means that the guide policy is providing feedback on the expected returns the learner partial generations will receive. Seminal imitation learning work like DAgger [2] shows that providing expert feedback on states generated by the learner is helpful for training a high quality policy. In this case, the guide policy is treated as an expert policy which the learner imitates and surpasses eventually.
>
> [1] Approximately Optimal Approximate Reinforcement Learning, Kakade and Langford 2002
>
> [2] A Reduction of Imitation Learning and Structured Prediction to No-Regret Online Learning, Ross et al. 2011
>
> __Theoretical Justification__
>
> Thank you for this critique. Due to the space limit, we mainly provided intuitive explanations of the benefits of using a guide policy. Please see the answer above where we provided more connections to the seminal works in the RL and IL literature. If that’s not helpful, could we ask the reviewer to provide more actionable details on how we could make the section more self-contained?
>
> __Performance Gain__
>
> Please see our updated results we posted along with the rebuttal for more results with TL;DR. For a quick summary, we show that when doing best-of-n sampling with our RL policies (PPO included), PPO++ shows greater performance improvement than PPO.
>
> __Ablation Mixing Parameter__
>
> Thank you for this recommendation. We did not find this parameter to be particularly sensitive in our experiments, but we agree that this would be a valuable addition to the paper. We will work to add this to the appendix in the final version.
>
> __Stronger Black Box Model such as LLama2/GPT4__
>
> We definitely agree with the reviewer that this would be very exciting future work since in principle, the use of Llama2 as $\pi_g$ is perfectly valid. Note that given the growing concern of data contamination in LLM evaluations [1], experimental design becomes more difficult when evaluating on popular benchmarks such as IMDB and TL;DR summarization which have not been explicitly defined as held-out datasets by models such as Llama2 [1].
>
> [1] Llama 2: Open Foundation and Fine-Tuned Chat Models, Touvron 2023
>
> __Setting $\beta$=1__
>
> We would like to clear up a potential misunderstanding. When the mixing parameter is set to 1, we mean that all trajectories are mixed. That is every trajectory has some amount of $\pi_g$ tokens as well as $\pi_{\theta}$ tokens. For example in AggreVaTeD, this means that every trajectory has rollins from $\pi_{\theta}$ and rollouts from $\pi_g$. When doing the policy update, we would then update with the samples from $\pi_{\theta}$
>
> __Missing Results: D2LOLS TL;DR__
>
> Thank you for pointing this out. We will work to add these results to the appendix for the camera ready. We felt that investigating the individual components of D2LOLS (PPO++ and AggreVaTeD) was more valuable given our limited computational budget.
>
> __Missing Related Work__
>
> Thank you for pointing out this interesting work. We definitely missed this and will add a discussion for this in the related works. The main difference between STG and RLGF here is the different mixing strategies employed. In RLGF, we obtain entire partial sequence feedbacks while in STG, a mixed trajectory may involve a single token being replaced.

---

> ### Comment · Area_Chair_h2oy · 2023-11-20
>
> Hello, reviewer. Please review the author's response. Does the response address your concern about motivation and theoretical justification?

---

### Official Review · Reviewer_Q8S4 · 2023-11-01

**Soundness:** 3 good
**Presentation:** 3 good
**Contribution:** 3 good
**Rating:** 5
**Confidence:** 4

**Summary:**

This work extends RL algorithms to allow them to interact with a dynamic black-box guide LLM and propose RL with guided feedback (RLGF). Experiments show that this method achieves higher performance than supervised learning (SL) and RL baselines.

**Strengths:**

1. The method looks novel and presents a good extension to PPO.
2. The theoretical justification look rigorous.

**Weaknesses:**

1. Experimental results on both the sentiment sentence generation and TLDR dataset do not show that the proposed methods can outperform PPO, let alone significantly.
2. I am wondering what is the difference between the LLM policy \pi_\{theta} and the guide policy \pi_g. It is said that \pi_g is the SFT+nucleus sampling. But in PPO, the LLM policy model should also be a fine-tuned LLM on some tasks. In this case, what is the difference between the two policy models?

**Questions:**

1. What is the size of the LLM policy model? GPT2-large or GPT2-medium? Have you tried larger LLM models?
2. What is the model for evaluating the output-perplexity?
3. Why didn't you use GPT4 to evaluate the win-rate? I don't think LLAMA2-13B-Chat is able to provide good and fair evaluation over model outputs for the win rate.

---

> ### Author Response · Authors · 2023-11-19
>
> Thank you for your review. We hope to address your concerns here
>
> __Experimental Results__
>
> Please see our updated results we posted along with the rebuttal for more results with TL;DR. For a quick summary, we show that when doing best-of-n sampling with our RL policies (PPO included), PPO++ shows greater performance improvement than PPO.
>
> __Clarification of Guide and Learn Policy__
>
> Yes, you are correct. In our experiments, both our starting PPO policy $\pi_{\theta}$ and our guide policy $\pi_g$ are initialized with $\pi_{SFT}$. However, it's important to note that while $\pi_g$ remains fixed as a black box policy, providing PPO with additional states through the nucleus sampling decoding strategy, PPO itself is not fixed and gathers samples using the softmax sampling decoding strategy.
>
> __What LLMs were used?__
>
> - IMDB: GPT2-base
> - CommonGen: T5-base
> - TL;DR: GPTJ (6B)
>
> We have the exact hugging-face model specifications in the implementation details in the appendix. Note that we aimed to try varying types and sizes of LLMs across our experimentation.
>
> __Output Perplexity__
>
> We computed Output Perplexity with GPT2-Large. We thank the reviewer for pointing this out and will make this clearer in the final version.
>
> __GPT4 Winrate__
>
> We did our best to use open-source models for the entirety of our work. Open-source models remain publicly available, guaranteeing the reproducibility of our experiments and results. Whereas closed source models’ accesses can be depreciated [1] at any time, making it impossible to reproduce our results.  We will release our models to the huggingface hub for public usage and for full reproducibility of our results. In some preliminary experiments we found the relative winrate trends between LLama2 70B and LLama2 13B to be aligned. Working within our computational budget, we found that using Llama2 70B was difficult to execute for every experiment.
>
> [1] https://openai.com/blog/gpt-4-api-general-availability

---

> ### Comment · Area_Chair_h2oy · 2023-12-02
>
> Hello, reviewer. Please review the authors' response. Does the response address your concern about performance?

---

> > ### Comment · Reviewer_Q8S4 · 2023-12-03
> > **Have read author response, no change of score needs to be made**
> >
> > I have read the author response and the updated paper again (mainly on the tables). I am not sure which tables show the updated results. By looking at table 1 and 2, the RM score of TLDR dataset only improves from 6.01 to 6.11 by comparing SFT+PPO with SFT+PPO++. And the numbers of the rows of SFT+PPO an SFT+PPO++ in table 1 are very close. So I don't think this updated version can show that PPO++ can outperform PPO significantly.
> >
> > Besides, in terms of the "GPT4 Winrate" part, in order to justify a metric based on a LLM, we need to make sure that the LLM judgement agrees with human judgement in a good amount of ratio. If we do not have quantitative evidence to show that LLAMA2-13B model can highly agrees with human judgement, we cannot use it as an evaluation metric. Furthermore, GPT4 has some arxiv version like GPT-0613, which is fixed for benchmark and calibration.

---

### Official Review · Reviewer_HK72 · 2023-11-01

**Soundness:** 3 good
**Presentation:** 3 good
**Contribution:** 2 fair
**Rating:** 6
**Confidence:** 4

**Summary:**

The paper systematically adapts known techniques from reinforcement learning, imitation learning and learning to search for the purpose of fine-tuning a large language model to maximize a set reward. Authors combine these techniques to formulate concrete algorithms (e.g. PPO++ and D^2LOLS) under the common umbrella of "reinforcement learning with guided feedback" (RLGF). In short, the main unifying trait of new algorithms is the use of "guide" policy, typically in the form of another LLM that can generate reasonable, but not necessarily optimal candidate sequences. Authors evaluate their algorithms on IMDB (sentiment), CommonGen and Reddit Summarization and compare against standard (for LLM community) baselines for RLHF, supervised fine-tuning and zero-shot prompting. The paper also contains sensitivity analysis and theoretical justification for some algorithmic choices.

**Strengths:**

1. The paper manages to combine a diverse set of ideas from prior RL research and formulate multiple algorithms within the 9 page limit, which is no small feat.

2. Authors conduct comprehensive evaluations with multiple realistic tasks and near-SoTA language models. The experiments are using standard llm fine-tuning best practices and report multiple seeds (in most cases). While this is not outstanding, many recent papers do not pass this bar, therefore it feels like a strength.

3. For a paper about so many different ideas and algorithms, this one is reasonably well written and easy to follow.

**Weaknesses:**

My main concern is the choice of baselines. While authors compare against SFT and basic PPO, prior research developed alternative algorithms for fine-tuning LLMs on human feedback that also claim superiority to PPO and SFT. Authors even cite some of those works in the paper. Some of those algorithms are: DPO[1], APA[2], P3O[3], SLiC-HF[4] though there may be more.

* [1] https://arxiv.org/abs/2305.18290
* [2] https://arxiv.org/abs/2306.02231
* [3] https://arxiv.org/abs/2310.00212 - note: this was published after the paper submission and authors should feel free to ignore it
* [4] https://arxiv.org/abs/2305.10425v1

These works adopt different means to learn from human feedback: some of them compatible to RLGF while others can only be used as competitors. I believe that the paper would be improved if, for each competitor, authors either compare against it in the experiments or prove that it has no chance of outperforming RLGF algorithms or, if authors claim that RLGF algorithms are orthogonal, demonstrate how it performs in combination with those approaches.

Another, less important concern is about the paper structure. Authors manage to cram multiple algorithms (PPO++, AggreVaTeD, LOLS in Appendix B, D2LOLS) and evaluate all of them within the few allowed pages. As a result, the paper fills "crammed" despite authors' considerable effort.
Perhaps it would be better to explore one or two of those algorithms **in more detail** and leave the rest to appendix? Though, I will understand if authors deliberately choose otherwise.

For the last (and least) of issues, the idea of guiding the search for optimal policy is technically not novel outside the LLM fine-tuning domain, and the same can be said about other ideas presented in the paper. To reiterate, it is not a "problem" to be solved, and a good practical adaptation of existing methods is valuable in and of itself.

**Questions:**

Minor typos:

> page 1:  RL-based methods which utilize reward signals outperforms on the task metric
 outperform (plural?)

---

> ### Author Response · Authors · 2023-11-19
>
> Thank you for taking the time for a thorough review. We hope to address your concerns below:
>
> __Choice of Baselines__
>
> We agree with the reviewer that many of the cited works are relevant literature in the RLHF space. We’d like to emphasize two points:
> 1. Our work focuses on investigating direct algorithmic improvements to PPO when using a reward signal
> 2. Our work extends beyond preference data for reward signals. For example, we tested on CommonGen which uses an automatic metric CiDer-D and SPICE as the reward function for the task. We aimed to show that given any reward signal, whether it be a preference reward model in RLHF tasks or an automatic metric such as CiDer-D, RLGF can better optimize than PPO.
>
> We agree that it would be interesting to explore combinations of RLGF with many of the other methods; however, we believe this is well beyond the scope of a single paper, especially given the reviewer’s observation that this investigation is already quite dense.
>
> __Paper Structure and Novelty__
>
> Thank you for the suggestions. We can definitely see the merit of focusing on one or two algorithms in detail. On the flip side, we felt that demonstrating the full combination of guide-policy and policy interactions in the policy gradient formulation would be a valuable contribution to the space of Deep RL algorithms for text generation. As the reviewer pointed out, many of these ideas were explored in different domains, but we would like to emphasize that most of them did not have a deep algorithm counterpart with this work being the first to unify all of these methods into a single, policy gradient implementation. Furthermore, by investigating all these combinations, we hope we convinced the readers of which rollin-rollout interaction paradigms are better suited for text-generation than others.
>
> On a separate note, many of the works cited were concurrent work to ours, with all of them being archival preprints or just announced Neurips acceptances at the time of submission.

---

> ### Comment · Area_Chair_h2oy · 2023-11-20
>
> Hello, reviewer. Please review the author's response. Does the response address your concern about the choice of baseline?

---

### Author Response · Authors · 2023-11-19
**Additional Results on TL;DR**

Thank you to all the reviewers for taking the time to thoroughly review our work. Here is a table with some additional results that we will refer to in each individual response.

| Algorithm | RM Score | Win Rate |
| ---- | ---- | ---- |
| PPO (n=8) | 6.20 | %57.53 |
| PPO++ (n=8) | 6.52 | %60.30 |
| AggreVaTeD (n=8) | 6.11 | %54.12 |

These additional results are Best-of-8 results with our trained policies for each respective algorithm.

---

### Meta-Review · Area_Chair_h2oy · 2023-12-02

**Metareview:**

** Strengths of the Paper:**

1. **Innovative Approach:** Integrating a guide LLM into RL training represents a novel approach.

2. **Theoretical Justification:** The paper includes a theoretical discussion which helps readers understand the rationale behind the proposed methods.


** Weaknesses of the Paper and Missing Elements:**

1. **Marginal Performance Gains:** The reviewers note that the performance improvements over existing methods like PPO are marginal. This raises questions about the practical significance of the proposed approach. In response, the authors provide more results on the TL;DR dataset.

2. **Comparative Analysis with Other Methods:** The paper could be strengthened by including comparisons with other recent methods claiming superiority over PPO.

3. **Motivation and Presentation:** The motivation behind the paper could be clearer, and the presentation of the theoretical justification could be improved to make it more accessible to readers.

**Justification For Why Not Higher Score:**

The recommendation to reject the paper is grounded in several key concerns:

1. **Marginal Performance Gains:** The paper's central drawback is the minimal improvement in performance that the proposed methods offer compared to PPO. The authors present results using a best-of-8 approach on the TL;DR dataset (without further clear explanation). This narrow focus on specific settings raises doubts about the method's broader applicability and generalizability.

2. **Insufficient Comparative Analysis:** The paper falls short in providing a thorough comparative analysis with other recent methodologies that also purport to surpass PPO.

**Justification For Why Not Lower Score:**

N/A

---

### Decision · Program_Chairs · 2024-01-16

Reject